# Full Endoscopic Treatment for a Fibrosis Complication after Psoas Abscess

**DOI:** 10.3390/jpm13071166

**Published:** 2023-07-20

**Authors:** Álvaro Dowling Montalva, Rui Nei de Araujo Santana Junior, Marcelo Molina

**Affiliations:** 1DWS Spine Clinic Center, CENTRO EL ALBA-Cam. El Alba 9500, Of. A402, Región Metropolitana, Las Condes 9550000, Chile; 2Department of Orthopaedic Surgery, Faculdade de Medicina de Ribeirão Preto (FMRP) da Universidade de São Paulo (USP), Ribeirão Preto 14040-900, Brazil; 3Spirituality and Pain Committee of the Brazilian Society for the Study of Pain (SBED) Health Technologies and Medical Education, São Paulo 04014-012, Brazil; raraujojrneuro@gmail.com; 4Bahiana School of Medicine and Public Health, Salvador 40290-000, Brazil; 5Instituto Traumatológico de Santiago, Clínica Alemana, Santiago 7560801, Chile

**Keywords:** psoas abscess, endoscopic, debridement

## Abstract

Background: Psoas abscess is a challenging disease that may sometimes lead to a devastating prognosis. Early diagnosis and treatment are mandatory for better results in their treatments and to avoid complications. Purpose: There is no article regarding a fibrosis treatment of the psoas muscle with a psoas abscess that is treated with full endoscopic debridement (FED). Study design: a case report and literature review. Result: we successfully treated this case, who suffered from psoas fibrosis with a clinical and MRI diagnosis, with full endoscopic debridement. Conclusions: FED is a viable alternative to open debridement for this rare complication of a psoas muscle abscess.

## 1. Introduction

Mynter first described psoas abscess in 1881 [1]. The psoas muscle is in the lower lumbar region and originates from T12 to L4 of the spine, and it extends through the pelvis to the femur. Psoas abscess can be divided into primary and secondary, based on the presence or absence of the previous condition [2]. The primary is more common in tropical and developed countries, while the secondary can be monomicrobial or polymicrobial [3]. In developed countries with a high socioeconomic index, psoas abscess is most frequently caused by *Staphylococcus aureus*, *Streptococcus*, *Escherichia coli* and *Mycobacterium tuberculosis* [3]. Magnetic resonance imaging (MRI) and imaging exams facilitate diagnosis and increase the incidence. Reports suggest that approximately 75% of patients experience significant delays in diagnosis, leading to multiple emergency department visits and postponed definitive diagnostic studies [1]. Treatment success depends on early diagnosis [3]. Lumbar spondylodiscitis is the main reason for secondary psoas abscess. Timely detection and proper treatment can help avoid complications and fatalities. The go-to treatment for psoas abscesses is broad-spectrum antibiotics and drainage through either percutaneous or open surgery. However, in the case of secondary psoas abscess, the mortality rate can reach up to 18.9%, even with these treatments [4].

Less than 30% of patients exhibit the classic triad of fever, limp, and back pain, which are the primary clinical symptoms [5]; usually, the most common symptoms are fever and back pain. Presenting symptoms are generally non-specific and may have features suggestive of other diagnoses, such as iliobursitis or retrocecal appendicitis, septic arthritis of the hip, or hip arthritis [6]. The fact that the PA is located in a retroperitoneal space, such as a retro fascial space, makes for a challenging ultrasound diagnosis that is often missed 40% of the time. On the other hand, CT and MRI are relatively easy; the preference for CT is a performance with I.V. contrast, especially in the case of a secondary abscess. CT enormously facilitates the guidance for drainage of the pus material, making it possible to take culture and sensitivity for therapeutic implications. MRI is commonly the gold standard mainly because it is an excellent tool to see the PA and other causes of low back pain. It is the best visualization of all images; some centers use gallium-97, but it is not always disposable. Some limitations of using imaging diagnosis (tomography and MRI) for PA in the early stage are that sometimes they are undetected by these methods. In a 2015 publication [7], Toshiko Takada et al. found in a retrospective review of 15 patients’ diagnoses of ten years that the overall sensitivity of plain CT, enhanced CT, and plain MRI for PA was 78%, 86%, and 88%, respectively. Sensitivity from days one to five after the onset of the symptoms was only 33%. After six days, sensitivity was 100%. Consequently, both methods can fail to diagnose in the early stage. Considering the high mortality rate, a delayed diagnosis causes delayed treatment and a poor prognosis.

There are different and various strategies for psoas abscess treatment: Drainage with tomography guidance, open surgery, and conservative treatment, including rest, antibiotics, and pain management—all this for approximately three months; the main problem with this treatment is the disability for a long period, including the complications in the elderly such as lung and heart malfunctions, which are not uncommon. FED is advantageous in elderly patients or those with multiple medical comorbidities and a high risk of undergoing general anesthesia [8]. The higher life expectancy in the general population increases the chance of spine infections [1]. Hernia discectomy is often performed using posterolateral percutaneous endoscopic techniques; however, this method can also be helpful for treating cases of psoas or lumbar ventral epidural abscess, including younger patients. By using this approach, we can directly reach the abscess and perform lavage and drainage in a less invasive way than traditional open surgery [9]. Currently, in many endoscopic spine centers, surgeons are now skilled at performing full endoscopic debridement due to its numerous benefits over open surgery, including that full endoscopic debridement can be performed with local anesthesia, the culture rate is higher, less post-op pain, and faster patient recovery. In the past decade, we have transitioned to using full endoscopic early treatment interventions for spine infections. This approach involves taking cultures for accurate diagnosis and providing better pain control to reduce hospitalization time for patients. Traditional anterior surgical open approaches with fusion and instrumentation are risky and aggressive operations for thoracolumbar infections with a high index of complications [10]. All of this makes full endoscopic debridement attractive for novice surgeons, and it is gaining popularity.

## 2. Case Report

A 69-year-old female came in with back pain due to spondylodiscitis. She was treated with antibiotics for 90 days without improvement. Worsening symptoms included malaise, low back pain, and weakness in the lower limbs. She had paresis of the left lower limb with muscle strength grade 3/5 in the L4 innervated motor groups and grade 4/5 in the hip flexors. Computed tomography of the lumbar spine was performed, showing a low attenuation lesion in the left psoas. Percutaneous needle aspiration and medical treatment with vancomycin and ceftriaxone were attempted at another facility before the patient was admitted to our service for surgical debridement of her spinal infection. A new MRI scan of the lumbar spine with contrast showed an oval, homogeneous paravertebral lesion in the left psoas muscle below the L4 transverse process with hyperintensity on T2-weighted images (Figure 1a). The images suggested a differential diagnosis of abscess, granuloma, or fibrosis. Surgical debridement with the FED technique was proposed to the medical team, and the patient consented to it.

## 3. Surgical Technique and Recovery

Based on our experience and the location of the infection of the psoas fibrosis, we accessed the Transforaminal approach (also called the modified extraforaminal approach, MEA). The operation was performed with local anesthesia and sedation (MAC) with the patient in the prone position on a radiolucent table. A third-generation endoscope system (Vertebris Stenosis, RIWOspine GmbH, Knittlingen, Germany) was used. This oval endoscope measures 9.3 × 7.4 mm in outer diameter and has a 20° optical angle with a 5.6 mm diameter working channel and a 117 mm working length. An articulating radiofrequency probe (Elliquence, Baldwin, New York, NY, USA) was used for coagulation and tissue ablation. The MRI identified the lesion as anterior to the distal half of the left L4 transverse process (Figure 1b,c). The planning of the surgery was like a traditional transforaminal approach (MEA). After addressing the foramen and visualizing the characteristics of the disc without infection or pus, the endoscope was tilted to reach the tip of the transverse process under fluoroscopic guidance and direct video endoscopic visualization, and we found the psoas lesion. The lesion contained no pus but was composed of fibrotic and granulomatous tissue (Figure 1d–f). A delicate resection was performed, with subsequent visualization of the lesion-free muscles (Figure 1g). The outside-in approach was more convenient; in this case, the patient improved clinically with significantly less pain. The motor strength returned rapidly, and the patient could walk on the same day of surgery. Postoperatively, there was a transitory weakness of hip flexion, possibly due to psoas manipulation for lesion resection. It improved within six weeks. Oswestry’s disability index was less than 10, and VAS was 2. A postoperative surveillance MRI of the lumbar spine was performed. It demonstrated the complete removal of the infectious lesion (Figure 1h). The histopathological evaluation confirmed fibrotic tissue. Cultures from intraoperative specimens were negative for fungi, tuberculosis, and anaerobic and aerobic cultures, which were all negatives. Two months after the full endoscopic debridement, the patient was asymptomatic.

## 4. Discussion

Psoas abscess primarily occurs due to hematogenic or lymphatic dissemination and tends to affect children or young adults [11]. Risk factors include diabetes, intravenous drug use, HIV infection, renal insufficiency, and other forms of immunosuppression [12]. A secondary abscess develops directly from a psoas muscle infection propagation in an adjacent structure [13]. In this case, risk factors include trauma in the inguinal, hip, or spine region [1,2]. FED suits elderly patients with multiple medical conditions [14]. In the thoracolumbar region, the advantage of small incisions and low blood loss is of particular appeal because it mitigates the risk of the traditional transpleural approach [6].

Our 69-year-old patient did not improve after 90 days of antibiotic treatment with vancomycin and ceftriaxone at another hospital [15]. Our patient’s cultures were also negative, and the best option was an early treatment with endoscopic drainage. We treated the left paravertebral psoas abscess below the distal half of the left L4 transverse process with a complete endoscopic debridement. This surgical method became increasingly popular as a less aggressive alternative to open surgery techniques. The direct video endoscopic visualization of painful pathology allows diagnosis and treatment in the same operation. It is mainly used in spine pathologies such as spinal stenosis, interbody fusions, disc herniation, medial branch rhizotomy, and infection. Deploying the endoscopic technology platform in treating a psoas abscess is novel, and to the authors, the best knowledge has not been reported.

A compartment-based classification of spinal infection was proposed: (1) anterior infection with discitis, spondylodiscitis, and psoas abscess, and (2) posterior infection with epidural and paraspinal abscess [16]. The authors suggest using the transforaminal approach (MEA) for anterior and the interlaminar approach for a posterior infection. In the case of spondylodiscitis, the inside-out approach—where the working cannula is placed inside the disc space—is a straightforward technique, particularly for novice spine surgeons. The outside-in transforaminal approach (MEA) appears most suitable for the psoas debridement. Some authors recommend removing the working cannula with a rotating motion after finishing the procedure within disc space to diminish the risk of spreading the infection into the psoas muscle.

In our experience, it is essential to choose a surgical approach by endoscopic surgery based on the location, including posterior and anterior infection. In cases with anterior compromise, including spondylodiscitis, discitis, psoas abscess, and posterior cases were paraspinal and epidural abscesses. Initially, the transforaminal approach (MEA) was used for disc pathology to reach anterior pathology easily; in this case, the transforaminal approach (MEA) allowed access anteriorly without destructing the posterior structure. One of the main problems described in the literature is paresthesia and mild paresis after this approach, which can apparently be caused by lumbar plexus or exiting root damage. It is difficult to identify the cause in our techniques to avoid neural injury; we carefully tilt the endoscope from the transverse process to the psoas muscle, and with direct visualization, we gently perforate and feel inside, and we dissect the fibrous and granulomatous tissue. To use the serum pump safely, do not apply pressure greater than 45 mmHg. A study conducted by Fan et al. [9] examined the risks and complications associated with percutaneous endoscopic transforaminal discectomy (PETD). Out of 738 patients, 72 (9.76%) experienced various complications. These complications included recurrence in 2.30% (17 of 738), persistent lumbosacral or lower extremity pain in 3.79% (28 of 738), a dural tear in 1.90% (14 of 738), incomplete decompression in 0.81% (6 of 738), surgical site infection in 0.41% (3 of 738), epidural hematoma in 0.27% (2 of 738), and intraoperative posterior neck pain in 0.27% (2 of 738).

In a Ching-Hsiao Yu study, 34 Patients were treated by FED. A total of 28 patients were treated for primary spondylodiscitis, and 15 had concomitant psoas abscesses [17]. There was one patient who had a post-op infection and another who had a mixed infection. A total of 26 patients were treated with FED only, and 8 had a prior instrumented fusion. These authors did not report any significant intraoperative complications. We could not find a case report of FED treatment of symptomatic psoas fibrosis after abscess. There are some considerations in our patients worth discussing. She complained of mild paresthesia in her left leg after surgery. This symptom may have been a sequela from manipulating the lumbosacral plexus within the psoas muscles. The exiting nerve root at the surgical level may have also been traumatized during the transforaminal approach (MEA) [18]. During surgical procedures, nerve damage can occur; unfortunately, it cannot be corrected through surgery. Therefore, taking preventative measures is the most effective treatment for this complication. Precise anatomical knowledge of endoscopic surgery and a focus on safety must be considered. It is crucial for surgeons to be cautious and gentle during surgery to avoid causing harm to the nerves. If nerve damage does occur, recovery can be a lengthy process, even if it is reversible. Depending on the extent and location of the nerve damage, various treatments such as medication and rehabilitation may need to be performed to alleviate symptoms.

Awake surgeries in the field of neurosurgery have been limited to craniotomies over the past two decades; spinal surgeons pushed for local anesthesia and sedation for many minimally invasive techniques, such as decompressions and discectomies. Our goal is to ease the financial burden for patients by offering reduced costs, faster recovery times, and better outcomes [19]. Some limitations are mental health conditions, requiring a short operation time, and being limited to patients who can tolerate a prone position. According to the authors, it is crucial for the anesthesiologist to have sufficient experience in this method. Regular training for the anesthesiology team can often make a significant difference in determining whether this type of procedure is feasible or not. In particular, it is highly advantageous to minimize surgical risks for PA.

It is challenging to judge an early full endoscopic intervention because some cases may be approachable by percutaneous drainage, echo guidance, or tomography. Still, the versatility of the endoscopic approach and the various advantages—like the local anesthesia; the possibility of excellent direct visualization of the structure and perfect sampling; the possibility of reaching foramen, disc, and psoas abscess; and treating the infection with adequate drainage and debridement—make a desirable alternative for spine infections. For the approach, PA authors recommend using a traditional foraminal approach to examine the disc and the peripheral structure of the foramen; then, as we already mentioned, carefully tilt the endoscope with C-arm guidance to the tip of the transverse process of the affected level and start the drainage of the infections and debridement. An alternative must be approached directly to the psoas abscess. By injecting Omnipaque R (iohexol) (HealthCare Inc, Masrlborough, MA, USA) into the lumbar disc, the passage of pus between the lumbar disc and the psoas infection can be visualized; however, this alone is insufficient for draining only the disc. To access the psoas abscess directly, the endoscope must be moved in the same manner as in a lumbar plexus block [20]. Yu reported that six of the patients who experienced failure in his study had more complex medical histories. Three of these patients were HIV-positive and had a history of heroin use, which put them in the immunocompromised group [21]. The other three had previously undergone surgery and had instrumentation in place, which resulted in postoperative infections. This study found that the failure rate for FED surgery was higher in these special groups of patients. In a study by Yang, a success rate of 65% was reported for FED treatment of instrumented lumbar spine infections in 20 patients. However, in another one of Yang’s reports, the infection control rate for instrumented patients was lower than for primary spondylodiscitis (65% versus 86%). It is important to inform patients and families of the potential for higher failure rates when dealing with these types of cases [22].

Since PA is a rare disease and limited data are available, more time is required to determine the effectiveness of the new procedure accurately; however, the primary reports indicate a positive outlook.

## 5. Conclusions

The development of an advanced endoscopic technology platform allows the spine surgeon to expand the traditional indications for the procedure from a herniated disc and spinal stenosis to more complex surgical indications. Endoscopic debridement of infected psoas muscles is another example. FED should be considered as another choice in treating spine infections and complications.

## Figures and Tables

**Figure 1 jpm-13-01166-f001:**
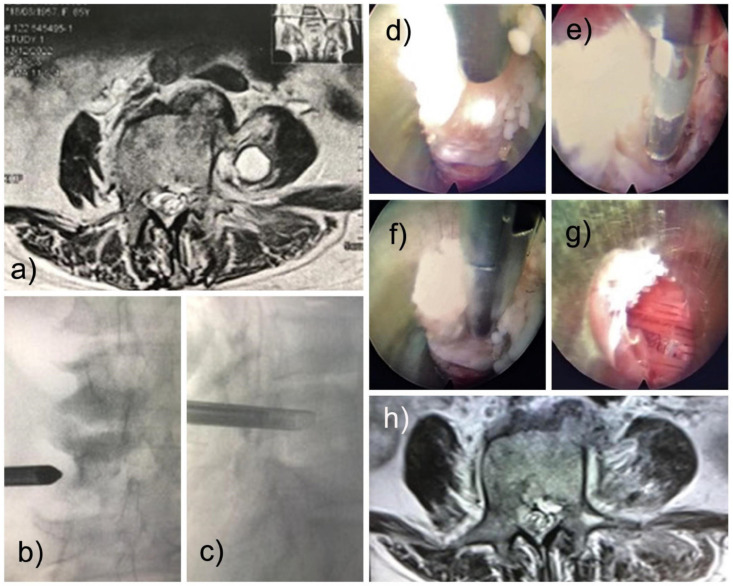
Shows a preoperative MRI scan of a 69-year-old female who initially presented back pain due to spondylodiscitis. The patient developed a psoas abscess shown as a round enhancing lesion on the T2-weighted axial MRI scan with intravenous contrast (**a**). The lesion was surgically treated with full endoscopic debridement (FED) with the working cannula placed in the center of the lesion below the left L4 transverse process (**b**,**c**). A fibrotic lesion was found intraoperatively (**d**,**e**), which was resected with the use of rongeurs (**f**) until the healthy red psoas muscle was visualized (**g**). A radiofrequency probe was used to achieve hemostasis. (**h**) A similar MRI scan was taken six weeks after the index FED confirmed the lesion’s successful removal. The patient recovered rapidly after surgery. The postoperative hip flexor weakness spontaneously resolved.

## Data Availability

The data presented in this study are available upon request from the corresponding author.

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
