# Peer review of "Full Endoscopic Treatment for a Fibrosis Complication after Psoas Abscess"

_jpm, 2023, doi:10.3390/jpm13071166_

Round 1

Reviewer 1 Report

Good case report which will expand the field of endoscopic spine surgery.

There needs to be substantial correction of grammar and English word usage. The content is good but the grammar and some usage is poor and makes it difficult to read.

Author Response

Dear Reviewer:

I appreciate your comments, and we strongly believe we made a much better case report this time. We tried to consider all the suggestions that you gave us, and we hope we succeeded in answering your concerns. We ordered the introduction which could have been clearer. It is also relevant to mention that even though the endoscopic approach has been used for Psoas abscess treatments, we could not find any other case of full endoscopic treatment for fibrosis after Psoas Abscess in the literature.

We're looking forward to hearing your new feedback.

Kind regards

Álvaro Dowling.

Reviewer 2 Report

The case report titled "Full Endoscopic Treatment for a Fibrosis Complication after Psoas Abscess" presents a novel approach to the treatment of psoas abscess adopted by the authors, consisting of a complete endoscopic debridement. The presented methods are innovative and can have a significant impact on surgical practice, particularly as an alternative to more aggressive open surgery techniques.The authors effectively present endoscopic technology as a novel, less invasive technique that allows for diagnosis and treatment during the same operation. This modern, less invasive method of treatment may find application not only in psoas abscess cases but also in other spine pathologies such as spinal stenosis, interbody fusions, disc herniation, medial branch rhizotomy, and infections. However, it's important to underscore that despite the attractiveness of the endoscopic approach, further research is needed to assess the long-term efficacy and safety of this technique. The authors should also discuss any potential limitations associated with the use of endoscopy in treating psoas abscesses. With this innovative technique, it would also be beneficial to see more data on clinical outcomes and potential complications. This would fully help understand the potential benefits and risks associated with the application of this technique. In conclusion, the authors effectively present the application of an endoscopic technique for the treatment of fibrosis complications following a psoas abscess. However, it is worth bearing in mind the need for further research to fully evaluate this method.

Fig.1 description :   ....healthy red - (not read)  psoas muscle.... 

Author Response

(The authors gave the same response as above.)

Reviewer 3 Report

There is already literature that treats psoas abscess with endoscopy, and there is nothing special about it. And I couldn't find a reason to take a transforaminal approach to treat psoas abscess.

There are dangerous structures such as the nerve root and radicular artery to access the transforaminal and go to the psaos, so a direct endoscopic approach is recommended.

And the patient is a patient who is being treated for spondylodiscitis, and it is not clearly described whether the treatment is accompanied by this

Author Response

(The authors gave the same response as above.)

Reviewer 4 Report

Thank you to the authors for putting their efforts into writing this article.

The article is well written and has a good structure.

However, major revisions are needed before the article is accepted.

First of all, the study design is not a case report and literature review. there is no systematization of the literature, I'd modify in case report and scoping review of the literature.

Furthermore there are some major revision:

1) in the introduction section,  I'd not say that Psoas muscle is in the LOWER lumbar spine. Indeed, it originates from T12 to L4, and you can find it also for lateral surgery for example in L2-L3.

2) In the intro section, it's not understandable what is present only in 30% of patients, please revise.

3) Why FED is advantageous only in elderly patients? that is correct but it's only the tip of the iceberg. In fact also in younger patients, FED could be a good alternative for surgical treatment, especially those ones that would require an open surgery. I think this could be stated with some references

4) In the intro section, there is a confusing architecture. The intro starts in talking about the pathology and its symptoms, then the Authors state about the FED treatment, then diagnostics and finally again FED treatment. I think the authors should organize better the intro.

5)in surgical tecnique and recovery, what is FEED? Moreover, int he discussion section what is a "mild parrhesia". I thinks these are typing errors

6) Did you perform an istological\microbiological analysis of the fibrotic nodule? It would be interesting to state these results as a completion of the case report

7) in the discussion section, the authors state that the mild postoperative paresthesia could be caused by the removal of the fibrotic nodule (which is correct in this reviewer opinion) and by the transforaminal approach. Transforaminal approach could cause irritation of the exiting root, but it is very rare the have paresthesia if the access with endoscope is cranio caudal and anteroposterior as possibile while reaching the foramen, due to the inclination of the exiting root. Furthermore, the authors did not analyze the role of the lesion itself on the emerging paresthesia. I'd revise this with some references.

8) In the intro section, the authors state that PA has a high mortality rate. This is not supported by references, and is not stated in the pathology description in the beginning of intro section. Please revise

Author Response

(The authors gave the same response as above.)

Round 2

Reviewer 3 Report

It is worth emphasizing that the content has undergone substantial enhancements, leading to significant improvement.I appreciate your explanation of the approach, which involves utilizing the transforaminal approach, accessing the extraforaminal region, and engaging with the psoas muscle. This method initiates within the foramen and follows the conventional extraforaminal approach through the transverse process. From what I understand, this approach might better be called the "Modified Extraforaminal Approach".

Author Response

Dear reviewer:

It's a pleasure hearing from you again and receiving your feedback and suggestions. We will certainly include them in the updated version. We will add this explanation when we refer to the "transforaminal approach" that can also be called the "modified extraforaminal approach" to be more precise with the concept and thus make it easier for future researchers to comprehend the article.

Kind regards

Álvaro Dowling.

Reviewer 4 Report

The authors have followed all the suggestions this reviewer suggested.

The paper can be accepted in this new form

Author Response

Dear reviewer:

It's a pleasure hearing from you again and receiving your feedback and suggestions. We will certainly include them in the updated version. 

Kind regards

Álvaro Dowling.